# Complete Response in Metastatic Clear Cell Renal Cell Carcinoma Patients Treated with Immune-Checkpoint Inhibitors: Remission or Healing? How to Improve Patients’ Outcomes?

**DOI:** 10.3390/cancers15030793

**Published:** 2023-01-27

**Authors:** Jonathan Thouvenin, Claire Masson, Philippe Boudier, Denis Maillet, Sabine Kuchler-Bopp, Philippe Barthélémy, Thierry Massfelder

**Affiliations:** 1Medical Oncology Department, Hospices Civils de Lyon, Hôpital Lyon Sud, 69310 Pierre-Bénite, France; 2Regenerative NanoMedicine, Centre de Recherche en Biomédecine de Strasbourg, Fédération de Médecine Translationnelle de Strasbourg (FMTS), UMR_S U1260 INSERM, University of Strasbourg, 67085 Strasbourg, France; 3Medical Oncology Department, Institut de Cancérologie Strasbourg Europe, 67200 Strasbourg, France

**Keywords:** clear-cell-renal-cell carcinoma, immune-checkpoint inhibitors, immunotherapy, complete response, first-line treatment

## Abstract

**Simple Summary:**

Immunotherapy-based combinations represent the front-line standard of care for metastatic clear cell renal cell carcinoma (mccRCC) patients. These combinations lead to an overall survival improvement and a dramatic increase of complete response rate raising the question of possible cure of mccRCC patients. This review summarizes the recent advances in RCC treatment and biological aspects underpinning a possible healing.

**Abstract:**

Renal-cell carcinoma (RCC) accounts for 2% of cancer diagnoses and deaths worldwide. Clear-cell RCCs represent the vast majority (85%) of kidney cancers and are considered morphologically and genetically as immunogenic tumors. Indeed, the RCC tumoral microenvironment comprises T cells and myeloid cells in an immunosuppressive state, providing an opportunity to restore their activity through immunotherapy. Standard first-line systemic treatment for metastatic patients includes immune-checkpoint inhibitors (ICIs) targeting PD1, in combination with either another ICI or with antiangiogenic targeted therapy. During the past few years, several combinations have been approved with an overall survival benefit and overall response rate that depend on the combination. Interestingly, some patients achieve prolonged complete responses, raising the question of whether these metastatic RCC patients can be cured. This review will focus on recent therapeutic advances in RCC and the clinical and biological aspects underpinning the potential for healing.

## 1. Introduction

Renal-cell carcinoma (RCC) represents approximately 2% of cancer diagnoses and deaths worldwide, with more than 430,000 cases and 179,000 deaths per year [1,2]. Almost 35% of patients present metastases at diagnosis and approximately 20–40% develop secondary metastases after nephrectomy [2,3].

Immune-checkpoint inhibitors (ICIs) in combination with either another ICI or with VEGFR (vascular endothelial growth factor receptor) antiangiogenic tyrosine kinase inhibitor (TKI) have dramatically improved metastatic renal-cell carcinoma (mRCC) patients’ outcomes in first-line settings compared to sunitinib. These combinations represent the new standard of care in the front-line treatment of mRCC patients according to the IMDC (International Metastatic RCC Database Consortium) criteria [4,5]. Indeed, the use of ICI in combination with ipilimumab (anti-CTLA-4, cytotoxic T lymphocyte antigen 4 monoclonal, antibody) and nivolumab (anti-PD1, programmed cell death receptor 1, monoclonal antibody), which was evaluated in the Checkmate 214 trial, was approved for intermediate- or high-IMDC-risk-group mRCC patients [6]. Furthermore, the combinations of ICI with anti-VEGF-TKI were approved for all IMDC-risk groups. 

These PD1-based combinations include pembrolizumab (anti-PD1 monoclonal antibody) + axitinib (anti-VEGFR TKI), which was evaluated in the KEYNOTE 426 trial [7], nivolumab + cabozantinib (anti-VEGFR, c-MET, AXL, c-KIT, and RET TKI), which was evaluated in the Checkmate9ER trial [8], and, finally, pembrolizumab + lenvatinib (VEGFR, PDGFR, FGFR, c-KIT, and RET TKI), which was evaluated in the CLEAR trial [9].

After a median follow-up ranging from 26.6 to 67.7 months in the different trials, all these combination therapies demonstrated significant overall survival (OS) improvement compared with sunitinib (Table 1). The complete response (CR) and objective response rate (ORR) were reported to have dramatically improved with these treatments. However, the optimal combination to use for each patient remains an unanswered question. While CR and ORR, assessed by an independent central review committee, improved by 12% and 39% with the ipilimumab–nivolumab combination [10], they improved by 10% and 60% with pembrolizumab–axitinib, 12% and 56% with nivolumab–cabozantinib and 16% and 71% with the pembrolizumab–lenvatinib combination, respectively. All the ORR and CR rates assessed either by central review committee or by investigators are listed in Table 1. 

Furthermore, the long-lasting responses achieved by patients either with only these systemic treatment combinations or with the addition of local therapy, such as surgery, radiotherapy, or interventional radiology techniques, raise the question of healing.

This review will focus on the clinical and biological aspects of the possible “healing” of mRCC patients thanks to immunotherapy-based combination therapies, the biomarkers associated with the responses, and strategies to improve the CR rate.

## 2. Definitions: Complete Response, Depth of Response, and Long Responders

The activity of ICIs is related to the activation of an immune response against tumoral cells. Therefore, unlike chemotherapy or targeted therapeutic agents, new patterns of response and durable response that persist even after treatment interruption are now described [14].

### 2.1. Complete and Partial Response

Complete response (CR) and partial response (PR) are defined, according to the RECIST 1.1 and iRECIST criteria, as the disappearance of all target lesions and as a decrease of at least 30% in the sum of the target lesions, respectively. With the use of ICIs, new patterns of responses have been integrated into the iRECIST criteria, such as pseudoprogression, hyperprogression, and dissociated response [15]. Pseudoprogression is defined by tumor shrinkage after an initial enlargement or the appearance of new lesions, hyperprogression refers to a rapid tumor progression after ICI treatment with no consensual definition, and dissociated response is defined by the persistence of growing lesions while others shrink [15]. Amongst the different trials evaluating ICIs based combination therapies in the first-line treatment of mRCC patients, there was some discrepancy between the ORR or CR rates assessed by local investigators and the IRRC (independent regulatory review commission). The reported data suggest that investigators significantly overestimate ORR compared to blinded reviewers [16]. Despite some differences between the populations included in each pivotal trial, the combination arms provided better ORR and CR compared with the sunitinib arm in each trial (Table 1). 

The differences in terms of the patients’ characteristics and the rates of prior nephrectomy in the reported trials could in part explain the CR-rate discrepancy between the ICI-based combinations (Table 1). 

Indeed, responses in primary sites may be harder to achieve compared to metastatic sites, possibly due to molecular heterogeneity [17]. In an exploratory post hoc analysis of the Checkmate 214 trial, the ipilimumab–nivolumab combination demonstrated a survival benefit as well as a renal-tumor reduction compared with sunitinib, even in mRCC patients without prior nephrectomy [18]. Indeed, 35% of patients achieved a primary-renal-tumor shrinkage of ≥30% in the nivolumab–ipilimumab arms, compared with 20% in the sunitinib arm, but no complete response was observed [18]. 

Discrepancies in ORR or CR rates assessed through independent review committees or by investigators may also explain the different reported data from each trial. In the Checkmate 214, the CR rate with the ipilimumab–nivolumab combination was 9% when assessed by IRRC compared to 11% by investigators [6]. 

Furthermore, the populations included in each trial were different. In Checkmate 214 and Checkmate 9ER [6,8], 23% and 22% of the patients had a favorable IMDC risk, respectively, while the risk was approximately 31% in Keynote 426 and CLEAR [7,9]. As IMDC patients with a favorable risk of IMDC respond better to TKI-based combinations, we can hypothesize that the ORR and CR rate may be influenced.

The number of metastatic sites at baseline was also different, with 73% of patients having more than two metastatic sites in Keynote 426, 71.5% in CLEAR, 80% in Checkmate 9ER, and 78% in Checkmate 214. We can hypothesize that a CR may be more difficult to achieve in patients with a significant tumor burden.

### 2.2. Depth of Response Correlates to Duration of Response

Depth of response (DepOR; maximum percentage of reduction from baseline in sum of target-lesion diameters), has shown to be prognostic for survival in mRCC in the TKI era [19]. Recently, it was demonstrated that DepOR is correlated with OS in mRCC patients treated with first-line ICI-based combination therapies. With ipilimumab–nivolumab, the data suggested that patients with a DepOR > 50% had the greatest OS benefit [20]. Similarly, data on the correlation between DepOR and OS in RCC patients treated with nivolumab–cabozantinib combination therapy were recently presented [21]. The DepOR results were based on the best overall response assessed by blinded independent central review and the best reduction threshold (CR and PR were subdivided in three subgroups based on tumor reduction: PR1, ≥80% and <100%; PR2, ≥60% and <80%; PR3, ≥30% and <60%, SD and PD, respectively). The progression-free survival (PFS) and OS according to the DepOR subgroups were analyzed after 6 months after the randomization landmark. Deeper responses with nivolumab–cabozantinib were associated with an improved 12-month PFS rate compared with sunitinib for CR (94.9% vs. 82.4%), PR1 (81.3% vs. 37.5%), and PR2 (72.1% vs. 53.2%). In both arms, increasingly deep responses led to better OS outcomes. The OS rates were comparable between arms for CR, PR1, PR2, and PR3 [21]. Therefore, it is logical to take into account to the depth of response as a possible endpoint in order to improve patients’ outcomes. 

### 2.3. Durable Response

Since ICIs allow the restoration of immune T-cell infiltration and stimulate anti-tumoral response, it is expected that the response to ICIs should be durable [22]. There is currently no consensual definition of a durable response or a long responder [14]. The authors of a meta-analysis of phase III trials evaluating immunotherapy defined a durable response as a patient having a PFS exceeding the median PFS of the whole population of patients treated with the same drug in the same trial by three times [23]. This definition avoids bias in terms of treatment and tumor type.

The 5-years follow-up of the Checkmate 214 study demonstrated durable benefits in the ipilimumab–nivolumab-combination population. The conditional estimates indicated that the majority of the patients who continued to respond at 3 years with the ICI combination continued to respond at 5 years [11]. The median duration of response (DOR) was longer (not reached vs. 24.8 months), with more ongoing responses in patients who received nivolumab–ipilimumab across risk groups (63% vs. 50%). Furthermore, more patients in this arm experienced a treatment-free interval without requiring subsequent systemic therapy compared with those treated with sunitinib (48% vs. 24%) [11]. The median DOR was also improved with the ICI and anti-VEGFR TKI combinations of pembrolizumab–axitinib (23.6 months vs. 15.3 months), nivolumab–cabozantinib (23.1 months vs. 15.1 months), and pembrolizumab–lenvatinib (25.8 months vs. 14.6 months), compared to sunitinib (Table 1). The DOR is a marker that should be taken into account in the choice of treatment in first-line settings. 

## 3. Clinical and Biological Problems in ICIs

In the previous TKI era, despite an interesting ORR, the CR rate remained very low, and patients experienced progression with the emergence of resistance. New agents were developed in order to counteract the mechanism of resistance; however, CR is only rarely achieved. Several mechanisms of resistance were described in the past: specific genetic mutations, as the genomic amplification of the therapeutic target; point mutations, leading to a reduction in the efficiency of the inhibitor; and the amplification and mutation of other genes, allowing tumor cells to maintain their oncogenic potential [24]. The development of resistance is clearly multifactorial not only in terms of genetic diversity, either preexisting or in development, but also in terms of intra-tumoral heterogeneity, although its role in therapy-induced resistance remains evasive [25]. 

Renal-cell carcinoma is an immunogenic malignancy, since (i) tumors are rich in lymphocytes infiltrates, (ii), spontaneous tumor regression may occur spontaneously, and (iii) earlier administration of immunotherapy (interleukin 2, interferon alpha) has proven effective in a subset of patients, with complete tumor regression observed at high doses despite, unfortunately, a significant rate of toxicities [26,27,28]. Obviously, all these facts suggest the potential therapeutic effect of the novel ICIs in mRCC. 

The treatment of mRCC has substantially improved since the advent of the new forms of immunotherapy with ICIs, as presented in the previous sections. Indeed, a significant proportion of patients achieve significant and durable responses. While this has clearly revolutionized treatments and improved outcomes for mRCC patients, a large subset of patients do not respond to ICIs. The same observations are made with combination regimens associating TKIs and ICIs in clinical trials and when used as first-line therapies [7,8,9], which is also reminiscent of the therapy-induced resistance to TKIs, as presented above. 

There is an urgent need to identify predictive clinical and/or biological biomarkers of efficacy or resistance that are associated with ICIs or ICI/TKI combinations, which would allow better clinical decision-making and ultimately lead to a more personalized approach. Different strategies have been assessed in order to improve the overall response rate, depth of response, and CR rate: the triplet approach (COSMIC 313), the adaptive approach (PDIGREE), acting on the microbiota, the biomarker-driven approach (Bionikk, the immune signature IMotion 151), and the multimodal approach.

### 3.1. Clinical Biomarkers of ICI Response

One of the major challenges of personalized oncology lies in identifying predictive biomarkers of response and resistance to therapy that are applicable in routine practice. Clearly, as presented in the previous sections, many new targeted or immune-based therapies, or new combination regimens, emerge on a regular basis for mRCC.

However, despite these new approaches, optimizing the treatment selection for each patient remains challenging. 

#### 3.1.1. IMDC Classification

The international metastatic-renal-cell carcinoma-database consortium (IMDC)’s prognostic model is used to predict the clinical outcomes of mRCC patients and contains six predefined factors (anemia, hypercalcemia, thrombocytosis, neutrophilia, Karnofsky performance status <80, and less than 1 year from diagnosis to first-line targeted therapy). It is an independent and effective predictive biomarker of poor OS in the first-line treatment of mRCC, and it can be applied to patients previously treated with targeted therapy [29]. It should be stressed that to date, the IDMC model remains the only prospectively validated predictive biomarker in mRCC. A post hoc analysis of the efficacy of the ipilimumab–nivolumab combination by the number of IMDC risk factors was conducted. The ORR with nivolumab–ipilimumab was consistent across 0 to 6 IMDC risk factors, whereas with sunitinib, it decreased with an increasing number of risk factors. The benefits of nivolumab–ipilimumab over sunitinib in terms of ORR (40–44% vs. 16–38%), OS (HR; 0.50–0.72), and PFS (HR; 0.44–0.86) were consistently observed in all groups of patients with intermediate-risk or poor-risk mRCC, regardless of the number of risk factors they had before starting treatment [30]. 

#### 3.1.2. PD-L1 Status

Various studies were published comparing the complete response rate (CRR), objective response rate (ORR), and PFS and/or progressive-disease rate (PDR), based on tumor PD-L1 status in mRCC patients; the results showed that tumor PD-L1 positivity in patients receiving ICIs was associated with better ORR and prolonged PFS, suggesting PD-L1 as a possible positive predictive clinical biomarker [31,32]. In these studies, the highest ORR and longest PFS were observed in PD-L1-positive patients receiving nivolumab plus ipilimumab. On the other hand, a subgroup analysis of the different TKI ICI combination trials suggest that PD-L1 positivity is associated with less significant benefits [7]. Thus, the immunohistochemical quantification of PD-L1 was developed as a promising biomarker for ICI response in mRCC in early clinical trials. 

However, (i) the use of various antibodies, (ii) the disparate cell populations analyzed, (iii) the different thresholds of positivity in the immunohistochemical evaluations, and (iv) a retrospective analysis of its expression clearly limited its use as a clinical biomarker of response in mRCC, as well as in other malignancies. 

Approximately 5% of patients with RCC present sarcomatoid features (sRCC) [33]. They have a poor prognosis, with targeted therapies showing limited efficacy [34]. The proportions of patients with sarcomatoid features were different across the different reported frontline PD-1-based combination studies (Table 1). A post hoc analysis of the Checkmate 214 trial evaluating the efficacy of ipilimumab–nivolumab vs. sunitinib in patients with metastatic sRCC was conducted. The rate of confirmed ORR was 60.8% with the combination versus 23.1% with sunitinib, with complete-response rates of 18.9% versus 3.1%, respectively. After 42 months of follow-up, the median OS favored ipilimumab–nivolumab (not reached (NR; 25.2–not estimable (NE)) versus sunitinib (14.2 months (9.3–22.9); *n* = 65; HR, 0.45 (95% CI, 0.3–0.7; *p* = 0.0004)). The magnitudes of the OS, PFS, and ORR benefits with nivolumab–ipilimumab observed in this patient subgroup were greater for those with tumor PD-L1 expression ≥1% versus those with tumor PD-L1 expression <1% [35].

#### 3.1.3. Neutrophil–Lymphocyte Ratio (NLR)

Beyond PD-L1, additional studies dealing with the identification of potential predictive clinical biomarkers were published. In two different retrospective studies, evaluating 42 and 65 mRCC patients treated with ICIs, a decrease in the neutrophil–lymphocyte ratio (NLR) level was shown to be associated with the clinical outcomes of ICIs, with a significantly better response to nivolumab [36,37]. The same observation was made on the C-reactive protein (CRP) level [38]. 

Recently, Yoshida et al. evaluated the potential role of eosinophilic features (that are linked to major immunological mechanisms) as predictive biomarkers in clear-cell RCC, in terms of the responses to TKIs but also to ICIs [38]. They included two different cohorts, one comprising 138 clear-cell RCC patients undergoing radical nephrectomy and the other 54 mRCC patients, both retrospectively. Three phenotypes were defined as clear, mixed, or eosinophilic type. The second cohort underwent biopsy, metastasectomy, or cytoreductive surgery before the initiation of systemic therapy, and included one group treated with TKIs (sunitinib, pazopanib, sorafenib, or axitinib) and another group treated with first-line ICIs or after TKIs (nivolumab or a combination of nivolumab–ipilimumab followed by nivolumab). They found a significant clinical benefit in the patients treated with ICIs compared with the patients treated with TKIs in the second group, in the mixed/eosinophilic types, suggesting that eosinophilic features are significantly correlated with ICI efficacy. 

Using a transcriptomic signature, Beuselinck et al. identified four robust ccrcc subgroups (ccrcc 1 to 4) that were associated with different responses to sunitinib treatment. The ccrcc1/ccrcc4 tumors were characterized by a stem-cell polycomb signature and CpG hypermethylation and had lower survival compared to the ccrcc3 tumors, which were sensitive to sunitinib and did not exhibit cellular responses to hypoxia, and to the ccrcc4 tumors, which exhibited sarcomatoid differentiation with a strong inflammatory, Th1-oriented but suppressive immune microenvironment, with a high expression of PD-1 and its ligands [39]. The BIONIKK trial aimed to evaluate the treatment efficacy and tolerability of nivolumab, nivolumab–ipilimumab, and VEGFR-TKIs in patients according to the tumor molecular groups defined by this signature [40]. The patients were randomly assigned to receive either nivolumab or nivolumab–ipilimumab in the ccrcc1 and ccrcc4 groups, or either a VEGFR-TKI or nivolumab–ipilimumab in the ccrcc2 and ccrcc3 groups. After a median follow-up of 18 months, in the ccrcc1 subgroup, the ORRs were 29% and 16% with nivolumab and nivolumab–ipilimumab, respectively, and 44% and 50% in the ccrcc4 group. In the ccrcc2 subgroup, the ORRs were 50% and 51% with TKI and nivolumab–ipilimumab, respectively, and no response was detected with TKI, compared with 20% for nivolumab–ipilimumab in the ccrcc3 group. This study demonstrates the positive effects of patient selection through tumor molecular phenotype on the selection of the most appropriate first-line treatment [40].

#### 3.1.4. Acting on the Microbiota

Very interestingly, the gut microbiome has emerged as an exciting field of research in oncology. Indeed, increasing evidence shows the interplay between the immune response and the gut microbiota, and its involvement in immune escape. The results obtained in preclinical mouse models and preliminary observations in limited patient series implicates the microbiome as a marker of response to ICIs [41,42]. Furthermore, it has been shown that fecal microbial transfer is able to convert ICI non-responders into responders [41], further indicating the therapeutic potential of the microbiota. 

To circumvent such limitations, the MITRE study was launched very recently. It is a large-scale prospective study currently recruiting patients with various cancer types, including melanoma, kidney cancer, or non-small-cell lung cancer, who are planned to receive standard ICIs [43]. The primary aim is to measure the ability of the microbiome signature to predict one-year PFS in patients with advanced disease, and the secondary aim is to measure, among others, the microbiome’s correlations with toxicity. It is expected from this large clinical study that the therapeutic manipulation of the microbiome will improve patients’ survival. Recently, a phase 1 trial evaluated whether the administration of CBM588, a bifidogenic live bacterial product, in mRCC treated with ipilimumab–nivolumab could improve immune response through the modulation of the gut microbiome. The PFS was significantly longer in patients receiving CBM588 (12.7 months versus 2.5 months, HR 0.15, 95% CI 0.05–0.47, *p* = 0.001). Although not statistically significant, the response rate was also higher in patients receiving CBM588 (58% versus 20%, *p* = 0.06) [44]. 

### 3.2. Multimodal-Therapy Approach

For oligometastatic or oligoprogressive disease, local treatments, including metastasectomy, radiotherapy or radio-interventional strategies, can be considered, and showed a benefit in terms of OS [5]. These decisions are usually discussed through a multidisciplinary tumor board. 

In the ICI era, the effect of local therapy on residual disease in order to improve CR rate remains uncertain. The optimal timing of cytoreductive nephrectomy, immediate or delayed, particularly in patients presenting CR or PR in response to ICI in metastatic sites, remain unknown. The question was recently evaluated in a French national retrospective study. Thirty mRCC patients who underwent partial or radical nephrectomy following a CR or PR after ICI-based treatment at the first- or later-line stage were included. At a median follow-up of 19.5 months after nephrectomy, 19 (63.3%) maintained a CR and discontinued systemic treatment [45]. The PFS rates were 96.7% at 12 months and 78.3% at 24 months. The OS rates were 100% at 12 months and 86.1% at 24 months [45]. The type of response at the metastatic site, for either PR or CR, was the only factor significantly associated with the risk of recurrence after nephrectomy. However, several questions remained unanswered, such as the benefits of surgery, particularly in the absence of a viable tumor in the nephrectomy specimen [45]. 

Using multimodal therapies in the treatment of mRCC patients, either metastasectomy in oligometastatic disease or the removal of residual disease after a good response to systemic therapies appear to be interesting strategies to improve survival while stopping systemic treatment.

### 3.3. Triplet or Adaptive Approach

Despite a great survival improvement for patients diagnosed with mRCC through ICI combinations, significant challenges remain. Novel ICI-based combination therapies are assessed in first-line settings, aiming to improve the CR rate. The results of the COSMIC-313 trial evaluating the combination of cabozantinib, ipilimumab, and nivolumab compared with placebo, ipilimumab, and nivolumab in previously untreated mRCC in patients with poor or intermediate risk of IMDC were recently presented at ESMO 2022. Eight hundred and fifty-five patients were randomized. The ORR was 43% in the triplet combination, with a CR rate of 3% in both arms, and 55% of the patients had a reduction of ≥30% in their lesions. The median PFS was not reached in the triplet arm compared to 11.3 months 95% CI (7.7–18.2), HR 0.73 (95% CI 0.57–0.94), *p* = 0.013 [45]. Adaptive strategies are also developed, such as in the PDIGREE trial (NCT03793166). Patients with mRCC started with ipilimumab–nivolumab, after which those who experienced CR continued nivolumab maintenance, those with PD switched to cabozantinib, and those with SD/PR were randomized between the addition of cabozantinib to nivolumab and the continuation of nivolumab maintenance [46].

### 3.4. New Therapeutic Agents 

The results of the combination of bempegaldesleukin, a pegylated IL-2 prodrug, with nivolumab compared sunitinib or cabozantinib in previously untreated mRCC patients were also presented at ESMO 2022. Six hundred and twenty-three patients were randomized. The trial was negative and after a median follow-up of 15.5 months, the ORR was 23% in the combination arm compared to 30.6% in the TKI arm [47]. Other trials are ongoing, including NCT02811861, the aim of which is to evaluate the combination of pembrolizumab, Lenvatinib, and belzutifan vs. pembrolozumab–quavonlimab, and lenvatinib vs. pembrolizumab and lenvatinib in mRCC patients in a first-line setting. 

Other strategies with experimental combinations of investigational agents (ICI and targeted therapies) are also ongoing, such as the phase 1–2 trial, MK-3475-03A (NCT04626479). 

The continuing development of new combination therapies and the identification of biomarkers predictive of the response to ICI will help to better select patients who will benefit of these treatments. Conventional ICIs lead to the restoration of T-cell activation and a reduction in T-cell depletion by specifically blocking PD-1, PD-L1, or CTLA-4. This will enhance the anti-tumoral immune response. As presented herein, it is obvious that these therapies have achieved some clinical efficacy in a subset of patients with mRCC. However, unfortunately, response rates and durability remain significant challenges. Consequently, it will be crucial in the near future to identify novel immune checkpoints and new combinations of therapeutic strategies. 

## 4. Conclusions and Perspectives

As presented in this review, several promising clinical and biological biomarkers have been identified, although without conclusive results to support their potential predictive value in mRCC. It is expected that future prospective, biomarker-driven studies will provide useful information about biomarkers that may be used to better predict responses to ICIs, especially given that a significant proportion of patients do not experience any benefits from ICI, ICI/ICI, or ICI/TJI combination therapies. As shown herein, other more promising clinical or biological entities than the early PD-L1 may potentially exist. Despite the complexity involved in identifying and validating such biomarkers, a composite biomarker will probably fulfill all the criteria. Thus, the same questions remain: (i) Which patients will best respond to and which patients will not respond to ICIs? (ii) How responses to ICI treatment be extended, maximized, and secured? How can therapeutic resistance be overcome?

## Figures and Tables

**Table 1 cancers-15-00793-t001:** Main results from large phase III trials in ccRCC first-line metastatic setting.

	Checkmate-214 [6,10,11]	Keynote-426 [12]	Checkmate-9ER [13]	CLEAR [9]
Treatment arms	Nivolumab–Ipilimumab (n = 550) Sunitinib (n = 546)	Pembrolizumab–Axitinib (n = 432) Sunitinib (n = 429)	Nivolumab–Cabozantinib (n = 323) Sunitinib (n = 328)	Pembrolizumab–Lenvatinib (n = 355) Sunitinib (n = 357)
Median follow-up, months	67.7	42.8	32.9	26.6
IMDC-risk status (Fav/Int/poor), %	Nivo–Ipi: 23/61/16 Sun: 23/61/16	Pembro–Axi: 32/55/13 Sun: 31/57/12	Nivo–Cabo: 23/58/19 Sun: 22/57/21	Pembro–Lenva: 31/59/9 Sun: 35/54/10
Metastatic site (Lung/Lymph node/Bone/Liver) %	Nivo–Ipi: 69/45/20/18 Sun: 68/49/22/20	Pembro–Axi: 72/46/24/15 Sun: 72/46/24/17	Nivo–Cabo: 74/40/24/23 Sun: 76/40/22/16	Pembro–Lenva: 70/48/24/17 Sun: 69/46/24/17
Previous nephrectomy	Nivo–Ipi: 82 Sun: 80	Pembro–Axi: 83 Sun: 83	Nivo–Cabo: 69 Sun: 71	Pembro–Lenva: 74 Sun: 73
Patients with sarcomatoid features in interm-/poor-risk patients, %	Nivo–Ipi: 17 Sun: 15	Pembro–Axi: 18 Sun:18	Nivo–Cabo: 11 Sun: 13	Pembro–Lenva: 8 Sun: 6
ORR/CR,% Central review	Nivo–Ipi: 42/9 Sun: 27/1	Pembro–Axi: 60/10 Sun: 40/3.5	Nivo–Cabo: 56/12 Sun: 28/5	Pembro–Lenva: 71/16 Sun: 36/4
ORR/CR,% Investigator review	Nivo–Ipi: 41/11 Sun: 28/1	NA	NA	Pembro–Lenva: 69/10 Sun: 34/7
Disease-control rate (PR + CR + SD), %	Nivo–Ipi: 75 Sun: 75	Pembro–Axi: 89 Sun: 83	Nivo–Cabo: 88 Sun: 69	Pembro–Lenva: 95 Sun: 86
Progressive disease (PD), %	Nivo–Ipi: 18 Sun: 14	Pembro–Axi: 11 Sun: 17	Nivo–Cabo: 6 Sun: 17	Pembro–Lenva: 5 Sun: 14
Median Duration of Response (95% CI), months	Nivo–Ipi: NR (49.5–NE) Sun: 23.7 (19.4–29)	Pembro–Axi: 23.6 (1.4–43.4) Sun: 15.3 (2.3–42.8)	Nivo–Cabo: 23.1 (20.2–27.9) Sun: 15.1 (9.9–20.5)	Pembro–Lenva: 25.8 (22.1–27.9) Sun: 14.6 (9.4–16.7)
Median PFS (95% CI), months and HR	Nivo–Ipi: 12.3 (9.7–16.5) Sun: 12.3 (9.8–15.2) HR 0.86 (0.73–1.01), *p* = 0.063	Pembro–Axi: 15.7 (13.6–20.2) Sun: 11.1 (8.9–12.5) HR 0.68 (0.58–0.80), *p* < 0.0001	Nivo–Cabo: 16.6 (12.819.8) Sun: 8.3 (7.0–9.7) HR 0.56 0.460.68), p significant	Pembro–Lenva: 23.9 (20.827.7) Sun: 9.2 (6.011.0) HR 0.39 (0.32–0.49), *p* < 0.0001
Median OS (95% CI), months and HR	Nivo–Ipi: 55.7 (46.3–64.6) Sun: 38.4 (32.0–44.0) HR 0.72 (0.62–0.85), *p* < 0.0001	Pembro–Axi: 45.7 (43.6–NE) Sun: 40.1 (34.3–44.2) HR 0.73 (0.60–0.88), *p* < 0.001	Nivo–Cabo: 37.7 (35.5–NE) Sun: 34.3 (29.0–NE) HR 0,70 (0.55–0.90), *p* significant	Pembro–Lenva: NR (NE) Su: NR (NE) HR 0.66 (0.49–0.86), *p* = 0.005

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
