# Peer review of "Complete Response in Metastatic Clear Cell Renal Cell Carcinoma Patients Treated with Immune-Checkpoint Inhibitors: Remission or Healing? How to Improve Patients’ Outcomes?"

_cancers, 2023, doi:10.3390/cancers15030793_

Round 1

Reviewer 1 Report

The data from the immune checkpoint inhibitor trials for the treatment of ccRCC showed that the drugs prolonged the period of being cancer free significantly. This review summarizes the recent advances in RCC treatment and immunotherapy-based combination therapies, the biomarkers associated with response and strategies to improve CR rate. There are still many problems.

1.    Some sentences are difficult to understand in English.

2.    The second part of Definitions, while helpful to the reader in defining some of the terms, is not appropriate as an important part of the review.

3.    The third part is the focus of this review. The content of 3.1 needs to be expanded.

Author Response

We thank the Reviewer for its appreciation of our work and for the comments that were raised.

The Reviewer raised 3 specific comments.

The first concerns the fact that some sentences are difficult to understand in English. We apologize for the embarrassment felt by the Reviewer when evaluating our work. After exchanges with one of the Section Manager Editor, we got the information that English Editors will help improve language quality if the manuscript is accepted. We hope that this answer will be satisfactory for the Reviewer. Thank you.

The second concerns the second part of the Definitions section of the original manuscript that was mentioned as not appropriate as an important part of the review. In fact, we believe that this Section is of importance for readers who may not be in the field but who may be interested in immune checkpoint inhibitors in cancer; However, as suggested, we have lightened this Section in the revised manuscript and move part of it to Section 3, and this is highlighted using the “Track Changes” function. We hope that these changes will satisfy the Reviewer. Again, thank you for this comment.

The third concerns the fact that chapter 3.1 needs to be expanded, and accordingly we have added data in the revised manuscript to also consider new data in the field.; this is highlighted using the “Track Changes” function. We hope that these additions will satisfy the Reviewer. Thank you for this comment.

Since we have restructured the text of the revised manuscript, we have modified the References Section accordingly, as well as in the text.

Overall, thank you very much for your comments that substantially improve the work presented herein.

Reviewer 2 Report

this is a comprehensive review about targeted treatment in RCC

Author Response

We thank Reviewer 2 for the kind appreciation of our work.

We are delighted that this work can be an important source of data for readers in the field.

This is a very encouraging opinion.

Reviewer 3 Report

The paper discusses the use of immunotherapy-based combinations as the standard of care for patients with metastatic clear cell renal cell carcinoma (mccRCC). These treatments have been shown to improve overall survival rates and increase the rate of complete responses, leading to the possibility of a cure for mccRCC patients. The review also highlights the immunogenic nature of RCC, with the presence of T cells and myeloid cells in an immunosuppressive state, which provides an opportunity for restoration through immunotherapy. Some patients have achieved prolonged complete responses, raising the question of the potential for a cure for these metastatic RCC patients. 

The papers describe the definitions and limitations of various response criteria for evaluating the effectiveness of immune checkpoint inhibitor (ICI) treatments in patients with metastatic renal cell carcinoma (mRCC). However, the writing is somewhat dense and difficult to follow. The paper could benefit from clearer organization and transitions between ideas.

 Additionally, the concept of "inter-observer variability" is mentioned, but not fully explained or elaborated upon. The paper also jumps around between different topics, including the use of ICIs, new response patterns, and differences between studies, without clearly connecting them.  Overall, some paragraphs lack cohesiveness and could benefit from more clear and thorough explanations.

There are other recent papers related not correctly referenced and updated. 

Author Response

We thank the Reviewer for is evaluation of our work and for the specific comments that were raised.

As for Reviewer 1, English was also a concern. Here also, we apologize for the embarrassment felt by the Reviewer when evaluating our work. After exchanges with one of the Section Manager Editor, we got the information that English Editors will help improve language quality if the manuscript is accepted. We hope that this answer will be satisfactory for the Reviewer. Thank you.

The Reviewer raised, in addition, 3 other concerns.

The first concerns the writing that was found dense and difficult to follow. We apologize fot his inconvenience felt by the Reviewer. As suggested, we have significantly improved the organization and the transitions between ides in the revised manuscript, and this is highlighted using the “Track Changes” function. We hope that reading will be easier to follow in the revised manuscript. Again, thank you for this comment.

The second concerns the "inter-observer variability" that we have mentioned in the original manuscript. We have now explained that in a more accurate way in the revised manuscript on pages 3, lines 94-101. We also improved the cohesiveness and comprehension of some paragraphs as requested by the Reviewer, and this is highlighted now throughout the text in the revised manuscript, using the “Track Changes” function. We hope that these changes will satisfy the Reviewer. Again, thank you for this comment.

The third concerns additional data with recent papers. Thank you for this comment. We have added new references 13 and 42 and rearranged all references, accordingly, considering also all modifications in the text, in the revised manuscript highlighted using the “Track Changes” function. It should be stressed that additional references were found but they did not fall into the scope of the Review. We hope that Reviewer will be satisfied with the data we added in the revised manuscript.

Overall, thank you for your comments that substantially improve the work presented herein.

Round 2

Reviewer 1 Report

I am satisfied by this last revised version of the manuscript.